# PROVABLE ROBUSTNESS BY GEOMETRIC REGULARIZATION OF ReLU NETWORKS

## ABSTRACT

Recent work has demonstrated that neural networks are vulnerable to small, adversarial perturbations of their input. In this paper, we propose an efficient regularization scheme inspired by convex geometry and barrier methods to improve the robustness of feedforward ReLU networks. Since such networks are piecewise linear, they partition the input space into polyhedral regions (polytopes). Our regularizer is designed to minimize the distance between training samples and the *analytical centers* of their respective polytopes so as to push points away from the boundaries. Our regularizer *provably* optimizes a lower bound on the necessary adversarial perturbation required to switch an example's label. The addition of a second regularizer that encourages linear decision boundaries improves robustness while avoiding over-regularization of the classifier. We demonstrate the robustness of our approach with respect to $\ell_\infty$ and $\ell_2$ adversarial perturbations on multiple datasets. Our method is competitive with state-of-the-art algorithms for learning robust networks while involving fewer hyperparameters. Moreover, applying our algorithm in conjunction with adversarial training boosts the robustness of classifiers even further.

## 1 INTRODUCTION

Neural networks have been very successful in tasks such as image classification and speech recognition. However, recent work (Szegedy et al., 2014; Goodfellow et al., 2015) has demonstrated that neural networks classifiers can be arbitrarily fooled by small, adversarially-chosen perturbations of their inputs. Notably, Su et al. (2017) demonstrated that neural network classifiers which can correctly classify "clean" images may be vulnerable to *targeted attacks*, e.g., misclassify those same images when only a single pixel is changed.

Previous work demonstrating this fragility of neural network classifiers to adversarial noise has motivated the development of many heuristic defenses including adversarial training (Madry et al., 2018) as well as certifiably robust classifiers such as randomized smoothing (Cohen et al., 2019; Salman et al., 2019) which characterize the robustness of a classifier according to its *smoothness*.

The intrinsic relationship between smoothness, or Lipschitz continuity—and their corresponding local variants—and robustness has motivated a variety of techniques to encourage uniform and local smoothness through the explicit regularization of approximations of the global and local Lipschitz constants (Zhang et al., 2018; Weng et al., 2018a;b). Recently, Lecuyer et al. (2019); Li et al. (2018); Cohen et al. (2019); Salman et al. (2019) proposed and extended a simple, scalable technique—randomized smoothing—to transform arbitrary functions (e.g. neural network classifiers) into certifiably and robust classifiers on $\ell_2$ perturbations.

Alternatively, previous work has also addressed adversarial robustness in the context of *piecewise-linear* classifiers (e.g., feedforward neural networks with ReLU activations). Wong & Kolter (2018); Jordan et al. (2019) propose to certify the robustness of a network $f$ at an example $x$ by considering a bound on the radius of the maximum $\ell_p$-norm ball contained within a union of polytopes over which $f$ predicts the same class. Related to our work, Croce et al.; Liu et al. (2020) propose maximum margin regularizers (MMR) which quantifies robustness of a network at a point according to the local region in which it lies and the distance to the classification boundary. Recent work also includes recovery and analysis of the piecewise linear function learned by an ReLU neural network during a training process (Arora et al., 2018; Montúfar et al., 2014; Croce & Hein, 2019). Typically, work in this area

centers around studying the complexity, interpretation, and improvement of stability and robustness of neural networks. For example, Montúfar et al. (2014); Serra et al. (2017) studied piecewise linear representations of neural networks and proposed the "activation profile" to characterize the linear regions.

In this work, we propose a novel regularizer for feedforward piecewise-linear neural networks, including convolutional neural networks, to increase their robustness to adversarial perturbations. Our Geometric Regularization (GR) method is based on the fact that ReLU networks define continuous piecewise affine functions and is inspired by classical techniques from convex geometry and linear programming. We provide a novel *robustness certificate* based on the local polytope geometry of a point and show that our regularizer provably maximizes this certificate. We evaluate the efficacy of our method on three datasets. Notably, our method works regardless of the perturbation model and relies on fewer hyperparameters compared with related approaches. We demonstrate that our regularization term leads to classifiers that are empirically robust and are comparable to the state of the art algorithms with respect to clean and robust test accuracy under $\ell_1$ and $\ell_\infty$-norm adversarial perturbations.

## 2  PRELIMINARIES

In this section, we briefly present background terminology pertaining to polytopes and their characterizations, adversarially robust classification, and the polytope decomposition of the domain induced by an ReLU network and its linearization over a given polytope.

### 2.1  PIECEWISE-LINEAR NETWORKS

An ReLU network is a neural network such that all nonlinear activations are ReLU functions, where we denote the ReLU activation by $\sigma : \mathbb{R} \to \mathbb{R}$, $\sigma(x) = \max\{0, x\}$. Informally, we define $\sigma : \mathbb{R}^d \to \mathbb{R}^d$ by $\sigma(x) = [\sigma(x_1), \dots, \sigma(x_d)]$. Let $f : \mathbb{R}^d \to [0, 1]^k$ be a feedforward ReLU network with $L$ hidden layers; for example, $f$ may map from a $d$-dimensional image to a $k$-dimensional vector corresponding to likelihoods for $k$ classes. Let $n_l$ be the number of hidden units at layer $l$, the input layer is of size $n_0 = d$, and let $W^{(l)} \in \mathbb{R}^{n_{l-1} \times n_l}$ and $b^{(l)} \in \mathbb{R}^{n_l}$ denote the weight matrix and bias vector at layer $l$, respectively. Since $f$ may be represented as the composition of $L + 1$ linear transformations and $L$ continuous piecewise-affine functions, $f$ must necessarily be continuous and piecewise-affine (for brevity, we will say piecewise-linear).

The half-space representation, or $\mathcal{H}$-representation, of convex polytopes is defined as follows:

**Definition 2.1** (Convex polytope)**.** A convex polytope $\mathcal{K}$ is the convex hull of finitely many points. Alternatively, a convex polytope may be expressed as an intersection of $m$ half-spaces. The $\mathcal{H}$-representation of a polytope is defined as the solution set to a system of linear inequalities $Ax \le b$:

$$\mathcal{K} = \{x : \forall j \in [m], a_j \cdot x \le b_j\}$$

Following definition 3.1 from Croce et al., a function is piecewise-linear if there exists a finite set of convex polytopes $\{Q_r\}_{r=1}^m$ (referred to as linear regions of $f$) such that $\cup_{r=1}^m Q_r = \mathbb{R}^d$ and $f$ is affine when restricted to each $Q_r$, i.e., can be expressed on $Q_r$ as $f(x) = Vx + a$.

Given a feedforward ReLU network $f$ and an input $x \in \mathbb{R}^d$, we intend to recover the polytope $Q$ conditioned on $x$ and the *linear restriction* of $f$ on $Q$. Therefore, we need to find $A$ and $b$ such that $Ax \le b$, where $A$ and $b$ define the intersection of $k$ half-spaces, i.e. a *polytope*, and $V$ and $a$ corresponding to the linearization of $f$ within this polytope such that $f(x) = Vx + a$ when restricted to $Q$.

We follow the formulation and notations of Croce et al.. If $x \in \mathbb{R}^d$ and $g^{(0)}(x) = x$ we recursively define the pre- and post-activation output of every layer:

$$f^{(l)}(x) = W^{(l)} g^{(l-1)}(x) + b^{(l)}$$
$$g^{(l-1)}(x) = \sigma(f^{(l)}(x))$$

The resulting classifier is then: $f^{(L+1)} = W^{(L+1)} g^{(L)}(x) + b^{(L+1)}$.

By rewriting $\sigma$ as an affine function, we can write $f^{(l)}$ formally as a composition of affine functions:

$$f^{(l)}(x) = W^{(l)}\Sigma^{(l-1)}(x)(\ldots(W^{(1)}x + b^{(1)})\ldots) + b^{(h)},$$

where we define $\Delta^{(l)}, \Sigma^{(l)} \in \mathbb{R}^{n_l \times n_l}$ conditioned on $x$ and defined elementwise as:

$$\Delta_{i,j}^{(l)} = \begin{cases} \text{sign}(f_i^{(l)}(x)) & \text{if } i = j \\ 0 & \text{otherwise} \end{cases}, \qquad \Sigma_{i,j}^{(l)} = \begin{cases} 1 & \text{if } i = j \text{ and } f_i^{(l)}(x) > 0 \\ 0 & \text{otherwise} \end{cases}$$

We now derive the polytope of $f$ at $x$ and the linear restriction of $f$ on polytope $Q$. By expanding the composition, we can concisely write $f^{(l)}(x) = V^{(l)}x + a^{(l)}$, where

$$
\begin{aligned}
V^{(l)} &= W^{(l)}(\prod_{h=1}^{l-l} \Sigma^{(l-h)}(x)W^{(l-h)}), \\
a^{(l)} &= b^{(l)} + \sum_{h=1}^{l-1}(\prod_{m=1}^{l-1} W^{(l+1-m)}\Sigma^{(l-m)}(x))b^{(h)},
\end{aligned}
\tag{1}
$$

and characterize the polytope $Q$ where $x$ lies as the intersection of $N = \sum_{l=1}^{L} n_l$ half-spaces:

$$\Gamma_{l,i} = \{z \in \mathbb{R}^d | \Delta^{(l)}(x)(V^{(l)}z + a^{(l)}) \geq 0\}.$$

## 2.2 POLYTOPE CENTERS

There are various definitions of the "center" of a polytope. Notably, the Chebyshev center of a polytope $\mathcal{K}$ is the center of the largest inscribed ball of $\mathcal{K}$, or equivalently, the interior point of $\mathcal{K}$ that maximizes the minimum distance between itself and the boundary of $\mathcal{K}$. More formally,

**Definition 2.2** (Chebyshev center). Let $\mathcal{K}$ be a convex polytope. The Chebyshev center of $\mathcal{K}$ with respect to an $\ell_p$ distance is the point $x \in \mathbb{R}^d$ which satisfies the following min-max problem:

$$\arg\min_{\hat{x}} \max_{x \in \mathcal{K}} ||x - \hat{x}||_p^2$$

Previous work explored the Chebyshev center in the context of adversarial machine learning (Croce et al.; Jordan et al., 2019). For example, Croce & Hein (2019) propose to include the minimum distance to the boundary in a non-smooth regularization term in their Maximum Margin Regularizer (MMR) to encourage samples to lie close to the Chebyshev centers of their respective polytopes.

In contrast, we explore the application of an alternative polytope center: the analytic center. The analytic center of a convex polytope $\mathcal{K}$ expressed via the $\mathcal{H}$-representation $Ax \leq b$ is canonically defined as an element of $\mathcal{K}$ that maximizes the product of the distances to the hyperplanes characterized by the rows of $A$ and the elements of $b$.

**Definition 2.3** (Analytic center). Let $\mathcal{K}$ be a convex polytope expressed via the $\mathcal{H}$-representation: $Ax \leq b$. The analytic center of $\mathcal{K}$ is the point $x \in \mathbb{R}^d$ which satisfies the following objectives:

$$x_{\text{ac}} = \arg\max_x \prod_{i=1}^{m}(b_i - \sum_{j=1}^{d} a_{ij}x_j) = \arg\min_x \left\{ -\sum_{i=1}^{m} \log(b_i - \sum_{j=1}^{d} a_{ij}x_j) \right\} \tag{2}$$
$$\text{s.t. } x \in \mathcal{K} \qquad\qquad\qquad \text{s.t. } x \in \mathcal{K},$$

where the second objective is canonically known as the *logarithmic barrier potential* (Nesterov & Nemirovskii, 1994).

It naturally follows that when boundary-planes of the polytope are symmetric about the analytic center (e.g. for polytopes that satisfy a *central symmetry* property), it exactly corresponds to the Chebyshev center. A polytope $\mathcal{K} \subset \mathbb{R}^d$ is *centrally symmetric* if $\mathcal{K} = -\mathcal{K}$; that is, $x \in \mathcal{K}$ if and only if there is a unique point $y$ such that the reflection of $x$ about $y$ is in $\mathcal{K}$: $2y - x \in \mathcal{K}$.

The analytic center, namely a weighted analytic center, has been extensively used in interior point (IP) methods for linear programming (Boyd & Vandenberghe, 2004). In general, the measure of closeness

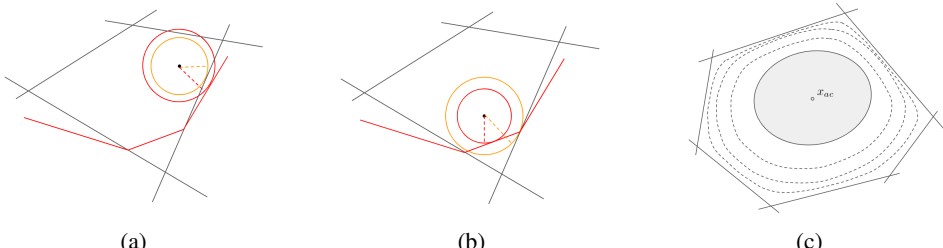

(a)            (b)            (c)

Figure 1: Lower bounds on the robustness of an input example $x$. **(a)** Case 1: the input $x$ (black) is closer to the boundary of the polytope $Q(x)$ (grey) than to the decision boundary (red). Our ACR term improves robustness in this case. **(b)** Case 2: the input $x$ is closer to the decision boundary than to the boundary of $Q(x)$. Smoothing $f$ addresses this case. **(c)** The analytic center $x_{ac}$ of a convex polytope. The inner ellipsoid corresponds to the *Dikin ellipsoid* at $x_{ac}$. The dashed lines show level curves of the logarithmic barrier function.

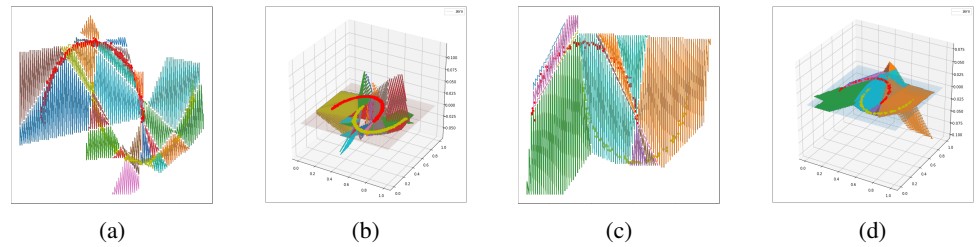

(a)         (b)         (c)         (d)

Figure 2: Recovered linear regions and linearizations of a trained feedforward, single hidden layer (128 neurons) ReLU network on the two-moons dataset. **(a)**: Polytopes defined by a vanilla network $f_{\text{vanilla}}$. **(b)**: Linearization of $f_{\text{vanilla}}$ restricted to the polytopes in (a). **(c)**: Polytopes defined by a robust network $f_{\text{robust}}$. **(d)**: Linearization of $f_{\text{robust}}$ restricted to the polytopes in (c). Note the difference in the granularity of the partitions and the overall smoothness of the classifier. The flat transparent pink (b) and grey (d) planes correspond to the plane defined by $y = 0$. The intersection of the networks with these planes correspond to the decision boundaries.

to the analytic center is an important factor for the convergence of center-following algorithms. The properties of the log barrier potential that make it a popular choice for interior point methods (including being twice continuously differentiable and convex) suggest we might use it as a regularizer to encourage robustness during learning.

Note, however, that the analytic center depends on how the set of particular inequalities is defined. The addition of redundant inequalities could feasibly push the analytic center arbitrarily close to the boundary of the polytope. We discuss this further in Sec. 3.

## 2.3 ROBUST CLASSIFICATION

Consider again the network $f : \mathbb{R}^d \to \mathbb{R}^k$, where the input might be a $d$-dimensional image and the output is a likelihood of that image belongs to one of $k$ classes. The associated classification is then $c(x) = \arg\max_{i \in [1,k]} f_i(x)$. In adversarial machine learning, we are not just concerned that the classification be correct, but we also want to be robust against adversarial examples, i.e. small perturbations to the input which may change the output to an incorrect class.

**Definition 2.4** ($\epsilon$-robust). $f$ is called $\epsilon$-robust with respect to norm $p$ at $x$ if the classification is consistent for a small ball of radius $\epsilon$ around $x$:

$$c(x + \delta) = c(x), \forall \delta : ||\delta||_p \leq \epsilon \tag{3}$$

This $\epsilon$-robustness is intimately related to the uniform and local Lipschitz smoothness of $f$. Recall that a function $f$ has finite, global Lipschitz constant $k > 0$ with respect to norm $|| \cdot ||$, if

$$\exists k \geq 0 \text{ s.t. } |f(x) - f(x')| \leq k \cdot ||x - x'||, \forall x, x' \in X$$

An immediate consequence of Eq. 3 and Def. 2.3 is that if $f$ is uniformly $L$-Lipschitz, then $f$ is $\epsilon$-robust at $x$ with $\epsilon = \frac{1}{2L}(P_a - P_b)$ where $P_a$ is the likelihood of the most likely outcome, and $P_b$ is the likelihood of the second most likely outcome (Salman et al., 2019). The piecewise linearity of ReLU networks facilitates the extension of this consequence to the *locally Lipschitz* regime such that for case 1 of Fig. 1, $L$ corresponds to the norm of the affine map characterized by $f$ conditioned on an input $x$.

Local smoothness is a property exhibited by provably and empirically robust networks. In Fig 2, the adversarially trained neural network is smoother compared to the vanilla network (larger linear regions with linear functions that have smaller slope and smoother transitions—"corner points"—between linear functions). In conjunction with smoothness, adversarial training also results in coarser partitions, i.e. larger polytopes, in contrast to vanilla networks. It is reasonable to assume that both of these properties are desirable in the context of robust classification.

In the case of piecewise linear networks, the lower-bound on the robustness of a network at a point given its associated polytope may be described with respect to two cases presented in Fig. 1(a)-(b). In case 1, the point may be closer to the boundary of its polytope compared to the decision boundary, and vica-versa for case 2. It follows that for case 1, the distance to the polytope boundary is a lower bound on the robustness of the network at that point. Croce et al. propose to optimize this certificate exactly, and we propose to optimize a relaxed version presented in the next section.

## 3 GEOMETRIC REGULARIZATION

### 3.1 ANALYTIC CENTER REGULARIZER (ACR)

We propose to adopt the logarithmic barrier potential as a regularizer to encourage points be near their analytical centers. We define the analytic center regularization (ACR) as

$$J_{\text{ACR}}(x) = \sum_{i=1}^{m} \log(a_i - \sum_{j=1}^{n} v_{ij} x_j) \tag{4}$$

where $V$ and $a$ are computed for network $f$ given an input $x$ via Eq. (1), and row-normalized. In related work, Croce et al.; Liu et al. (2020) both propose variants to the vanilla MMR formulation which involve regularizing a summation over the distances to the $k$ nearest boundary hyperplanes as opposed to only the distance to the nearest plane. This extension aims to ensure that the input is far from the boundary of the polytope. However, in addition to tuning this number, both methods employ a warm-up scheme to gradually increase the number of hyperplanes considered. In contrast, by regularizing the distance to the analytic center, ACR natively inherits the desirable properties of MMR while taking into account *all* hyperplanes comprising the boundary and involving no additional hyperparameters.

We note that many assumptions made regarding the analytic center rely on the polytope $P$ being *minimally* represented (e.g. the uniqueness of the analytic center). To elucidate this concept, we define redundant hyperplanes:

**Definition 3.1** (Redundant hyperplanes). Let $P := \{x | Ax \leq b\}$ be a polytope comprised of $m$ constraints and $i \in [m]$. $a_i x \leq b$ is a **redundant constraint**, or a **redundant hyperplane**, if it can be removed from the system without changing $P$, and **irredundant** otherwise. If all constraints in $Ax \leq b$ are irredundant, then the system is **irredundant**, and **redundant** otherwise.

In other words, the $\mathcal{H}$-representation of a polytope is not unique. In fact, there are infinitely many $\mathcal{H}$-representations of a polytope, but only one *minimal* representation. It is well-known that the inclusion of redundant hyperplanes may significantly affect the location of the analytic center.

**Lemma 3.1** (Boyd & Vandenberghe (2004)). *Let $P := \{x | Ax \leq b\}$ be an $\mathcal{H}$-representation of a polytope. The inclusion of redundant hyperplanes can make any interior point the analytic center.*

To address this, we propose a masked variant of the ACR term which ensures that only boundary hyperplanes of $P$ are included in the objective. However, recovering an irredundant $\mathcal{H}$-representation is nontrivial. We apply an approximate method which leverages the primal-dual equivalence between the half-space intersection problem and the convex hull problem. Details are provided in Appendix E.

In practice, we found that the application of ACR to all constraints (i.e. without removing redundant constraints) performs as well or better than than the masked variant in addition to having lower computational overhead. We hypothesize that this is due to the composite nature of neural networks and the fact that the output of the network $f$ is dependant on *all* hyperplanes—not just irredundant hyperplanes—and that ignoring redundant hyperplanes during regularization may adversely affect learning. Additional justification for inclusion of the redundant constraints when evaluating GR can be found in Fig. 11 where we see that the network trained with GR exhibits a more even and symmetric distribution of hyperplanes compared to other networks.

## 3.2 Linear Decision Boundary Regularizer (LDR)

Recent work (Moosavi-Dezfooli et al., 2019; Qin et al., 2019) has demonstrated that regularizing the curvature of the loss corresponds to regularizing the curvature of the decision boundary, thus improving robustness. Specifically, Moosavi-Dezfooli et al. (2019); Qin et al. (2019) respectively provide a tight certificate that is linear in the maximum eigenvalue of the Hessian of the loss with respect to the input and an upper bound on the adversarial loss. However, the regularization schemes proposed suffer drawbacks. Namely, both methods require the careful setting of a parameter which characterizes the local approximation of the Hessian and Qin et al. (2019) requires multiple iterations per training iteration. We exploit the intrinsic manifold structure of the data to encourage the loss to behave linearly around data via the following regularization term:

$$\int_{\mathcal{M}} ||\nabla_{\mathcal{M}}^2 f(x)||^2 d\mu(x). \tag{5}$$

In general, we cannot compute this integral because $\mathcal{M}$ is not known analytically. Belkin et al. (2006); Belkin & Niyogi (2008) propose the following discrete approximation that converges to the integral in the sample limit:

$$J_{\text{LDR}}(x) = \frac{1}{N_b^2} \sum_{\substack{i,j \in [N_b] \\ i \neq j}} \frac{||\nabla_x \ell(x_i) - \nabla_x \ell(x_j)||_2^2}{||x_i - x_j||_2^2}, \tag{6}$$

where $N_b$ is the minibatch size. Note that the contents of batches are randomized each epoch. The application of regularizers based on the graph Laplacian have also been explored in the past (Mishne et al., 2019), including in the context of adversarial robustness (Jin & Rinard, 2020). However, we propose to apply manifold regularization to the gradient of the loss instead of the weights of classifier. In the case of piecewise linear networks, this regularizer may be interpreted as reducing the angle between adjacent linear restrictions of $f$. Such sharp angles are known to facilitate brittleness in the presence of adversarial noise (Roth et al., 2019).

## 3.3 Geometric regularization of ReLU networks

Following the definitions above, we propose the following objective for adversarially robust training

$$L(\mathcal{D}) = \mathbb{E}_{\mathcal{D}}[\ell(x) + \lambda_A J_{\text{ACR}}(x) + \lambda_L J_{\text{LDR}}(x)] \tag{7}$$

where $\ell(x)$ is an arbitrary classification loss and $\lambda_A$ and $\lambda_L$ are hyper-parameters to be optimized. We note that, given an input $x$, a forward pass through $f$ is sufficient to compute both the linearization of $f$ around $x$ and the associated region (the polytope). Given these entries, optimizing the regularization term corresponds to solving a smooth, unconstrained convex problem - which can be done very efficiently using gradient descent. Analogous to adversarial training, LDR requires computation of $\nabla_x \ell(x)$ – performed via backpropagation. However, the gradients are computed only once per batch as opposed to iteratively as in adversarial training-based methods.

## 3.4 Geometric Robustness Guarantees for ReLU networks

We show that GR provably optimizes a lower bound on the radius of robustness. Given an input $x$ and network $f$, let $d_B(x)$ be the distance from $x$ to the boundary of its polytope and let $d_D(x)$ be the distance from $x$ to the decision boundary defined by $f$. To recover robustness guarantees when $d_B(x) \leq d_D(x)$, and demonstrate that GR optimizes a lower bound on the robustness, we rely on the *Dikin Ellipsoid* (DE), defined by the Hessian of the logarithmic barrier function (Nesterov & Nemirovskii, 1994).

**Definition 3.2** (Dikin ellipsoid). The Dikin ellipsoid of radius $r$ of a polytope $\mathcal{K}$ expressed by $Ax \leq b$ centered at $\hat{x}$ is defined as the set: $\mathcal{E}_r(\hat{x}) = \{x | (x - \hat{x})^T H(\hat{x})(x - \hat{x}) \leq r\}$, where $H$ is the Hessian of the log-barrier function defined to be $H = A^T S^{-2} A$ with $s_{ij} = \begin{cases} a_i x - b_i & i = j \\ 0 & \text{otherwise} \end{cases}$.

Notably, the Hessian of the logarithmic barrier function and the Dikin ellipsoid describe well the local geometry of $\mathcal{K}$ at $x$ in the following sense (we refer to Nesterov & Nemirovskii (1994) for details and proofs): (1) the Dikin ellipsoid is contained inside $\mathcal{K}$: $\mathcal{E}_1(x) \subseteq \mathcal{K}$ for any $x \in \mathcal{K}$. (2) The Hessian of the log-barrier changes slowly with respect to its local norm; while exploding to infinity at the boundary of $\mathcal{K}$. In other words, the gradient of the logarithmic barrier function at points near the boundary of $\mathcal{K}$ is dominated by a component with direction perpendicular to the boundary and pointing towards the interior of $\mathcal{K}$. (3) The volume of the Dikin ellipsoid is guaranteed to be at least $\frac{1}{m} Vol(\mathcal{K})$ where $m$ is the number of hyperplanes comprising the boundary of $\mathcal{K}$. An example of an analytic center and its Dikin ellipse are given in Figure 1c.

The properties of the Dikin ellipsoid imply that it may be used to construct a certificate at $x$ by taking the length of the shortest semi-axis of the ellipsoid. In particular, the length of the minor axis of the Dikin ellipsoid at $x$ may serve as a uniform lower-bound on the radius of robustness, and optimizing Eq. 4 provably improves this radius. More concretely, we get the following robustness guarantees:

**Theorem 1** (ACR Robustness guarantee for ReLU networks)**.**

1. *If $d_B(x) \leq d_D(x)$, then $f$ is locally robust at $x$ with radius $\epsilon = 1/\sqrt{\lambda_{max}}$ where $\lambda_{max}$ is the maximum eigenvalue of the Hessian matrix (and $1/\sqrt{\lambda_{max}}$ is the length of the minor axis of the Dikin ellipsoid) of the log-barrier potential function (maximized when $J_{ACR}$ is minimized).*

2. *If $d_B(x) \geq d_D(x)$, let $g = \nabla L_x(x)$. Then, for some constant $c$, the radius $\epsilon = \frac{c}{||g||} - \frac{2\nu c^2}{||g||^3}$ where $\nu$ is the maximum eigenvalue of the Hessian of $L(\mathcal{D})$(maximized as $J_{LDR}$ is minimized).*

Note that $i$-th axis of the ellipse described by the quadratic form $x^T A x = r$, corresponds to the $i$-th eigenvector of $A$ with length $\frac{1}{\sqrt{\lambda_i}}$.

As previously stated, we propose to use the length of the shortest minor sub-axis as a certificate of robustness. We call this certificate the *Dikin certificate*. The proof that the Dikin certificate is a lower bound on the radius of robustness is given in the appendix, and relies on the following property and lemma which directly follow from the definitions of the Dikin ellipsoid and the analytic center.

**Property 3.1.** *Let $\mathcal{K}$ be a convex polytope. Let $x$ be an interior point of $\mathcal{K}$ and let $\mathcal{E}_x$ be the Dikin ellipse defined at $x$. Then $\mathcal{E}_x \subseteq \mathcal{K}$.*

**Lemma 3.2.** *Let $\mathcal{K}$ be a centrally symmetric convex polytope, and let $H$ be the Hessian of the logarithmic barrier function defined on an irreducible $\mathcal{H}$-representation of $\mathcal{K}$. Then, $x_{ac} = x_{cheb}$ and $\lambda_{max}(H(x_{ac})) \leq \lambda_{max}(H(x)) \ \forall x \in \mathcal{K}$.*

In other words, the point that satisfies the largest radius of robustness with respect to the Dikin certificate is exactly the analytic center.

## 4 EXPERIMENTS

### 4.1 MAIN EXPERIMENTS

We provide a variety of experiments to demonstrate the performance of GR. We evaluate GR on three datasets: MNIST (LeCun & Cortes, 2010), Fashion MNIST (Xiao et al., 2017), and CIFAR-10 (Krizhevsky et al.). We consider robustness with respect to both $\ell_2$ and $\ell_\infty$ distances. We use three criteria: upper and lower bounds on the robust test error for a given threshold $\epsilon$ and the mean lower bound on the pointwise radius of robustness. Lower bounds on the robust test error for $\ell_2$ and $\ell_\infty$ are computed using the attack via Projected Gradient Descent (PGD) (Madry et al., 2018). Upper bounds and robust radii are computed using Wong & Kolter (2018).

We report our results in Table 1. We compare seven methods: plain training, adversarial training (AT) of (Madry et al., 2018), the robust loss of (Wong & Kolter, 2018) (KW), the MMR and MMR +

Table 1: Results for 7 training schemes: plain, at (adversarial training), KW, MMR and MMR+AT, GR, GR+AT, and 5 evaluation schemes for ReLU networks: clean test error (TE), lower (LB) and upper (UB) bounds on the robust test error via Wong & Kolter (2018), and average radius of robustness as estimated by Wong & Kolter (2018) (KW). The robustness radii are computed on the first 1000 points of the respective test sets. Models evaluated according to the available code.

| | CNN | | | | | | | | FC1 | | | | | | | |
|---|---|---|---|---|---|---|---|---|---|---|---|---|---|---|---|---|
| | perturbation: $\ell_\infty$ | | | | perturbation: $\ell_2$ | | | | perturbation: $\ell_\infty$ | | | | perturbation: $\ell_2$ | | | |
| | TE | LB | UB | KW | TE | LB | UB | KW | TE | LB | UB | KW | TE | LB | UB | KW |
| **MNIST** | $\epsilon = 0.1$ | | | | $\epsilon = 0.3$ | | | | $\epsilon = 0.1$ | | | | $\epsilon = 0.3$ | | | |
| plain | 0.92 | 78.32 | 100 | 0.0 | 0.85 | 3.1 | 100 | 0.04 | 1.56 | 98.32 | 100 | 0.0 | 1.73 | 9.7 | 66.3 | 0.217 |
| plain + AT | 0.98 | 12.1 | 100 | 0.008 | 0.87 | 1.8 | 100 | 0.14 | 1.56 | 7.10 | 99.0 | 0.002 | 1.15 | 2.6 | 16.9 | 0.255 |
| KW | 1.21 | 3.03 | 4.4 | 0.096 | 1.11 | 2.2 | 6.0 | 0.32 | 2.26 | 8.59 | 10.9 | 0.087 | 1.19 | 2.4 | 5.2 | 0.703 |
| MMR | 2.39 | 11.9 | 6.0 | 0.088 | 2.57 | 5.8 | 11.6 | 0.38 | 2.1 | 22.5 | 24.9 | 0.066 | 2.40 | 5.9 | 8.8 | 0.548 |
| MMR + AT | 1.24 | 5.6 | 3.6 | 0.091 | 2.12 | 4.6 | 9.7 | 0.38 | 2.04 | 14.0 | 14.1 | 0.082 | 1.77 | 3.8 | 6.4 | 0.592 |
| GR | 2.14 | 10.13 | 5.8 | 0.087 | – | 4.4 | 12.1 | 0.38 | 1.81 | 17.5 | 26.2 | 0.063 | – | 3.5 | 8.2 | 0.563 |
| GR + AT | 2.12 | 8.70 | 3.9 | 0.089 | – | 4.2 | 10.9 | 0.38 | 2.05 | 13.71 | 21.3 | 0.081 | – | 3.1 | 5.9 | 0.261 |
| **F-MNIST** | $\epsilon = 0.1$ | | | | $\epsilon = 0.3$ | | | | $\epsilon = 0.1$ | | | | $\epsilon = 0.3$ | | | |
| plain | 9.38 | 98.42 | 100 | 0.0 | 9.32 | 57.1 | 100 | 0.03 | 9.48 | 100 | 100 | 0.0 | 9.70 | 42.8 | 91.8 | 0.18 |
| plain + AT | 13.62 | 29.8 | 73.0 | 0.0 | 8.10 | 20.4 | 100 | 0.07 | 13.0 | 31.3 | 95.5 | 0.008 | 9.15 | 19.9 | 61.4 | 0.11 |
| KW | 21.31 | 32.8 | 32.4 | 0.046 | 13.08 | 18.5 | 21.7 | 0.17 | 21.31 | 32.8 | 32.8 | 0.034 | 11.24 | 17.2 | 22.7 | 0.46 |
| MMR | 17.57 | 34.2 | 33.6 | 0.033 | 12.85 | 25.4 | 35.3 | 0.17 | 18.11 | 37.6 | 42.0 | 0.022 | 13.28 | 25.0 | 28.0 | 0.50 |
| MMR + AT | 15.84 | 31.3 | 30.7 | 0.041 | 13.42 | 26.2 | 39.1 | 0.21 | 15.84 | 31.3 | 34.8 | 0.027 | 12.12 | 19.7 | 23.4 | 0.66 |
| GR | 17.54 | 35.13 | 34.5 | 0.031 | – | 25.1 | 35.6 | 0.15 | 16.38 | 34.91 | 43.7 | 0.021 | – | 24.1 | 28.1 | 0.47 |
| GR + AT | 15.76 | 31.41 | 32.1 | 0.039 | – | 26.0 | 40.2 | 0.19 | 15.99 | 33.78 | 36.2 | 0.026 | – | 17.6 | 24.0 | 0.64 |
| **CIFAR10** | $\epsilon = 2/55$ | | | | $\epsilon = 0.1$ | | | | | | | | | | | |
| plain | 24.63 | 91.0 | 100 | 0.0 | 23.31 | 47.2 | 100 | 0.02 | | | | | | | | |
| plain + AT | 27.04 | 52.5 | 88.5 | 0.001 | 25.82 | 35.8 | 100 | 0.04 | | | | | | | | |
| KW | 39.27 | 48.16 | 48.0 | 0.004 | 40.24 | 43.9 | 49.0 | 0.16 | | | | | | | | |
| MMR | 34.59 | 57.17 | 61.0 | 0.002 | 40.92 | 50.6 | 57.1 | 0.16 | | | | | | | | |
| MMR + AT | 34.36 | 49.27 | 54.2 | 0.003 | 37.75 | 43.9 | 53.3 | 0.13 | | | | | | | | |
| GR | 38.91 | 57.79 | 63.9 | 0.002 | – | 52.39 | 58.3 | 0.11 | | | | | | | | |
| GR + AT | 37.36 | 48.34 | 56.1 | 0.003 | – | 41.95 | 54.2 | 0.09 | | | | | | | | |

Table 2: Comparisons to state-of-the-art first and second-order robust regularizers. TRADES (Zhang et al., 2019), Locally Linear Regularization (LLR) (Qin et al., 2019), TULIP (Gradient Regularization) (Finlay & Oberman, 2019), and Curvature Regularization (CURE) (Moosavi-Dezfooli et al., 2019). We evaluate each approach using the CNN architecture described in Table 3 and utilize the PGD $\ell_\infty$ attach with $\epsilon = 0.1$ for MNIST and $\epsilon = \frac{2}{255}$ for CIFAR10.

| | MNIST | | CIFAR10 | |
|---|---|---|---|---|
| | TE | LB | TE | LB |
| GR (this work) | 2.14 | 10.13 | 38.91 | 57.79 |
| LLR | 1.13 | 10.43 | 25.83 | 58.14 |
| TULIP | 2.03 | 12.74 | 31.23 | 59.85 |
| TRADES ($\beta = 1$) | 0.97 | 5.70 | 27.01 | 53.39 |
| CURE | 1.07 | 11.34 | 26.8 | 61.74 |

with adversarial training scheme (Croce et al.), and our approach (GR) with and without adversarial training. All schemes are evaluated on a one hidden layer fully connected network with 1024 hidden units (FC1) and a convolutional network (CNN) with 2 convolutional and 2 dense layers. Note that unlike other reported results which fine-tune their defense to each attack type, we demonstrate empirical robustness against multiple perturbation types by training *one* model per architecture and dataset and report its robustness against different attack types. Additional details are in the Appendix.

On MNIST, we outperform or match MMR's clean test error and lower-bound robust accuracy performance with respect to both $\ell_2$ and $\ell_\infty$-norm perturbations. With respect to robust accuracy, we note that although the GR certificate is a lower bound on the MMR certificate, we closely match MMR's KW-based average certified radius. On F-MNIST and CIFAR10, we again match MMR with respect to clean test error and empirical robust accuracy and certifiable radius. We emphasize that Croce et al. perform grid search on model parameters for each dataset *and* perturbation, while our performance is based on a single model trained per-dataset.

We compare our approach to state-of-the-art first and second-order regularizers in Table 2. Details of these algorithms are provided in the appendix. We note that TRADES and AT remain the state-of-the-art heuristic methods, while GR performs competitively with other provable methods that encourage local linearity of the loss. We note that these methods (i.e. LLR, TULIP, and CURE) necessitate

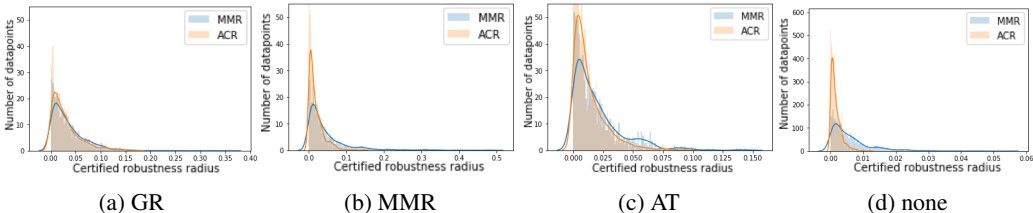

| (a) GR | (b) MMR | (c) AT | (d) none |

Figure 3: Distributions of MMR and GR robustness certificates for regularized networks. For networks trained with GR and AT, the GR certification matches closely with MMR.

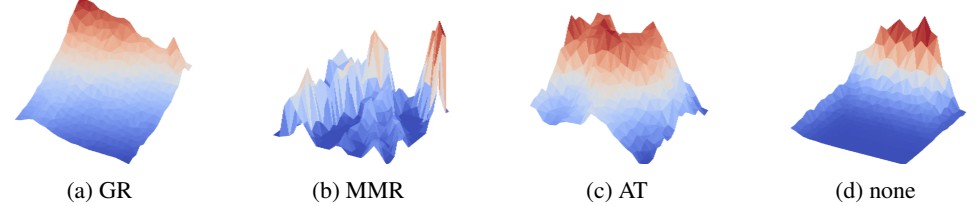

| (a) GR | (b) MMR | (c) AT | (d) none |

Figure 4: Local loss surface near decision boundary of networks trained on the two-moons dataset.

iterating or sampling around each training example to compute their regularization terms and, and thus require additional overhead and setting hyperparameters governing their sampling procedures.

As mentioned before, the GR certificate is a lower bound on the certificate proposed by Croce et al. which is a lower-bound on the radius of robustness. We present the empirical differences in point-wise certified radii in Fig. 3. For networks that are adversarially trained or regularized with GR, the distribution of point-wise radii of robustness is close. In contrast, the distributions are significantly different for vanilla networks and networks trained with MMR. The implication is that for certain networks and inputs, the Dikin ellipsoid may not optimally characterize the polytopes, in particular, if the $\mathcal{H}$-representation of a polytope exhibits redundancy concentrated on one side, or if the polytope is poorly conditioned in the sense that $H$ is poorly conditioned ($\mathcal{K}$ is geometrically "elongated").

### 4.2 ROBUSTNESS TO GRADIENT OBFUSCATION

Athalye et al. (2018); Qin et al. (2019) demonstrated that empirical robustness of neural networks may often be due to the effect of gradient obfuscation—i.e. the network learns to fool first-order attacks by contorting the loss surface. Although this non-linearity of the loss surface may cause networks to appear to exhibit robustness to adversarial attacks, Athalye et al. (2018) further demonstrated that merely running PGD for more iterations reliably produces adversarial perturbations. In contrast, networks that exhibit robustness while being close to locally linear exhibit "true" robustness to adversarial attacks. In Fig 4, we demonstrate that our regularizer produces models that exhibit such robustness to gradient obfuscation. We plot local regions of the loss surface near the decision boundary for various networks and show that training with GR leads to a smooth loss in the region of samples near the decision boundary, replicating the effect of adversarial training, while the local loss surface exhibited by networks trained with MMR leads to highly non-linear loss surfaces.

### 5 CONCLUSION

We have introduced a method based on leveraging the geometry of piecewise-linear neural networks for training classifiers that are provably robust to norm-bounded adversarial attacks. Our approach is based on the notion that piecewise-linear classifiers with large linear regions and smoother decision boundaries are robust. We demonstrate that our method learns robust networks that exhibit properties typical to robust networks and perform competitively with current state-of-the art methods, including related, geometrically-motivated techniques. By exploiting the geometry of the network and data, our method relies on fewer hyperparameters compared to related approaches. Furthermore, our method is scalable and leads to models which are simultaneously robust to multiple perturbation models.

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

## A  Preliminaries

Let $Ax \leq b$ be an irreducible, affine matrix inequality characterizing a polytope in $\mathbb{R}^d$ where $A \in \mathbb{R}^{m \times n}$, $x \in \mathbb{R}^n$ and $b \in \mathbb{R}^n$. We denote the feasible set characterizing the polytope by $\mathcal{K}$:

$$\mathcal{K} = \{x \in \mathbb{R}^m | Ax - b \leq 0\}$$

Without loss of generality, let the rows of $A$ be normalized under any $\ell_p$ norm. Then, the $\ell_p$ distance between any interior point $x$ of $\mathcal{K}$ to the $i$-th plane on the boundary of $\mathcal{K}$ is nothing but $d_i(x) = a_i^T x - b_i$.

For completeness, we restate the curvature-based certificate of Moosavi-Dezfooli et al. (2019) below.

**Theorem 2** (Moosavi-Dezfooli et al. (2019)). *Let $\ell(x)$ denote the loss of a neural network $f$ evaluated at $x$, and let $r^*$ be be the minimal perturbation necessary to fool the classifier. Let $H$ be the Hessian matrix of $\ell(x)$, $H = \frac{\partial^2 \ell}{\partial x_i \partial x_j} \in \mathbb{R}^{d \times d}$. Then, $||r^*||$ is the radius of robustness of $f$ at $x$ with respect to the second order Taylor expansion of $\ell$ at $x$:*

$$r^* = \arg\min_r ||r|| \text{ s.t. } \ell(x) + \nabla\ell(x)^T r + \frac{1}{2} r^T H r \geq t$$

*for a threshold $t$. Let $x$ be such that $c := t - \ell(x) \geq 0$, and let $g = \nabla\ell(x)$. Assume that $\nu := \lambda_{\max}$ and let $u$ be the eigenvector corresponding to $\nu$. Then, we have*

$$\frac{||g||}{\nu}\left(\sqrt{1 + \frac{2\nu c}{||g||^2}} - 1\right) \leq ||r^*||)$$

$$\leq \frac{|g^T u|}{\nu}\left(\sqrt{1 + \frac{2\nu c}{(g^T u)^2}} - 1\right)$$

*The above bounds can be further simplified to:*

$$\frac{c}{||g||} - 2\nu\frac{c^2}{||g||^3} \leq ||r^*|| \leq \frac{c}{g^T u}$$

## B  Proofs

### B.1  The Dikin ellipsoid of radius 1 is contained within $\mathcal{K}$. (Boyd & Vandenberghe, 2004)

Let $\mathcal{E}_x$ denote the Dikin ellipse of radius 1 at $x \in \mathcal{K}$. Note that since $x \in \mathcal{K}$, $d_i(x) \geq 0 \forall i$. Let $y \in \mathcal{E}_x$. Additionally, let $H$ be the Hessian of the logarithmic barrier function at a feasible point $x$:

$$H(x) = A^T S^{-2} A \text{ with } s_{ij} = \begin{cases} d_i(x) & i = j \\ 0 & \text{otherwise*} \end{cases}$$

The statement can be proven by showing that $y \in \mathcal{K}$ (i.e. $y$ is feasible). Equivalently, we will show that $d_i(y) \geq 0 \ \forall i$.

$$\begin{aligned}
(y - x)^T H(x)(y - x) &\leq 1 \quad \text{by definition} \\
\implies \sum_{i=1}^m \frac{\langle a_i, y - x\rangle^2}{d_i^2} &\leq 1 \\
\implies \frac{\langle a_i, y - x\rangle^2}{d_i(x)^2} &\leq 1 \quad \forall i \quad \text{since all entries in the summation are nonnegative} \\
\implies \left(\frac{d_i(x) - d_i(y)}{d_i(x)}\right)^2 &\leq 1 \quad \forall i \\
\implies \left|1 - \frac{d_i(y)}{d_i(x)}\right| &\leq 1 \quad \forall i
\end{aligned}$$

Hence, for all $i$, $0 \leq \frac{d_i(y)}{d_i(x)}$, so $d_i(y) \geq 0 \ \forall i$, and $y \in \mathcal{K}$.

## B.2 LEMMA 3.2

Recall the problem of recovering the Analytic center defined in Eq 4:

$$x_{\text{ac}} = \arg\max_{x \in \mathcal{K}} \prod_{i=1}^{m} \left( b_i - \sum_{j=1}^{d} a_{ij}x_j \right) = \arg\max_{x \in \mathcal{K}} \prod_{i=1}^{m} d_i(x) \tag{8}$$

For brevity, let $F = \prod_{i=1}^{m} d_i(x)$. Then the problem may be concisely written as

$$\arg\max_{x \in \mathcal{K}} F(x) \tag{9}$$

We will show that for any convex polytope $\mathcal{K}$ satisfying a central symmetry property (for example, *zonotopes*), that the maximizer of this problem is precisely a unique *origin* of the polytope. By definition, for centrally symmetric polytopes, for any constraint $a_i^T x \leq b$ there is a corresponding constraint $a_i^T x \leq -b$, and there exists a unique interior point $x_c$ such that if $x \in \mathcal{K}$, its inversion with respect to $x_c$ is also in $\mathcal{K}$: $\bar{x} \in \mathcal{K}$ where $\bar{x} = 2x_c - x$. We call this point $x_c$, the *origin* of $\mathcal{K}$. A simple consequence of this definition is that $x_c$ lies at the midpoint of the line $\overline{x\bar{x}}$. It follows that if $x_i$ is the minimum-distance projection from $x$ onto boundary plane $i$, and if $\bar{x}_i$ lies on plane $j$, then $d_i = d_j$.

The lemma is a simple consequence of this definition. For any perturbation applied to $x_c$, let $\epsilon_i > 0$ be the variation in the $\ell_p$ distance associated with the perturbation to boundary plane $i$. By the linearity of the boundary planes, the variation in the $\ell_p$ distance suffered by the inverse boundary plane of boundary plane $i$ is $-\epsilon_i$.

$$F(x) = \prod_{i}^{m/2} (d_i(x_c) - \epsilon_i)(d_i(x_c) + \epsilon_i)$$

$$\leq \prod_{i}^{m/2} d_i(x_c)^2 = F(x_c)$$

so $\arg\max_{x \in \mathcal{K}} F(x) = x_c$, and $x_{\text{ac}} = x_c$. Accordingly, $x_c$ also corresponds to the Chebyshev center. Note that we can restate the definition of the Chebyshev center using the above notation:

$$x_{\text{cheb}} = \arg\max_{x} \min_{i} d_i(x) \tag{10}$$

Again, for any perturbation to $x_c$, we have that

$$\min_{i}\{\min\{d_i(x_c) - \epsilon_i, d_i(x_c) + \epsilon_i\}\} = \min_{i}\{d_i(x_c) - \epsilon_i\}$$

$$\leq \min_{i} d_i(x_c)$$

so the solution to Eq. 10 is the origin of $\mathcal{K}$, and $x_{\text{ac}} = x_{\text{cheb}}$.

Finally, let $H$ be the Hessian of the logarithmic barrier function at a feasible point $x$:

$$H(x) = A^T S^{-2} A \text{ with } s_{ij} = \begin{cases} d_i(x) & i = j \\ 0 & \text{otherwise*} \end{cases}$$

Then, for any interior point $x$, it follows that

$$\lambda_{\max}(H(x_{\text{ac}})) = \max_{i} \frac{1}{\sqrt{d_i(x_{\text{ac}})}}$$

$$\leq \max_{i} \frac{1}{\sqrt{d_i(x)}} = \lambda_{\max}(H(x))$$

## B.3 THEOREM 1

### PROOF OF (1)

First, note that if $d_B(x) \leq d_D(x)$, then the predicted class of $x$ does not change for ball of radius $d_B(x)$ around $x$—$B_p(x, d_B(x))$. Let $\mathcal{E}_x$ be the Dikin ellipsoid at $x$ characterized by $H(x)$, the

Hessian of the logarithmic barrier potential evaluated at $x$. Let $r_i$ be the length of $i$-th shortest sub-axis of $\mathcal{E}_x$. By definition,

$$r_i = \frac{1}{\sqrt{\lambda_i}}$$

where $\lambda_i$ is the $i$-th largest eigenvalue of $H(x)$.

Since $\mathcal{E}_x \subseteq \mathcal{K}_x$, the polytope in which $x$ lies, by Property 3.1, it follows that

$$\min_i r_i = \frac{1}{\sqrt{\lambda_{\max}}} \leq d_B(x)$$

where $\lambda_{\max}$ is the largest eigenvalue of $H(x)$ and $\frac{1}{\sqrt{\lambda_{\max}}}$ is a lower bound on the minimal $\ell_p$-norm perturbation necessary to change the class.

PROOF OF (2)

Note that if $d_D(x) \leq d_B(x)$, then the Dikin certificate is not a sufficient lower-bound on the radius of robustness $r^*$, however, the bound of Theorem 2 holds globally:

$$\frac{c}{||g||} - 2\nu \frac{c^2}{||g||^3} \leq ||r^*||$$

where $c$ is a loss-dependant constant, $g = \nabla \ell(x)$, and $\nu = \lambda_{\max}$.

## C  MAIN EXPERIMENTS

Table 3: Architectures for main experiments for number of classes $nc$.

| FC1 | CNN |
|---|---|
| FC(1024) | Conv(16, $4 \times 4$, 2) |
| ReLU | ReLU |
| FC($nc$) | Conv(32, $4 \times 4$, 2) |
| | ReLU |
| | FC(100) |
| | ReLU |
| | FC($nc$) |

Table 4: Ablation experiments. We evaluate networks trained using both ACR and LDR terms, ACR only, and LDR only on $\ell_\infty$ perturbations. We see that the best performance is achieved when both terms are used.

| | MNIST | | FMNIST | | CIFAR10 | |
|---|---|---|---|---|---|---|
| | TE | LB | TE | LB | TE | LB |
| Both | 1.81 | 17.5 | 16.38 | 34.91 | 37.36 | 57.79 |
| LDR | 1.95 | 15.8 | 21.0 | 46.83 | 35.85 | 61.87 |
| ACR | 2.6 | 52.6 | 16.01 | 73.8 | 36.21 | 76.49 |

We abbreviate one hidden layer fully connected network with 1024 hidden units with FC1. The convolutional architecture that we use is identical to that of Wong & Kolter (2018); Croce et al. —consisting of two convolutional layers with 16 and 32 filters of size $4 \times 4$ and stride 2, followed by a fully connected layer with 100 hidden units. For all experiments we use batch size 128 and we train the models for 100 epochs. Moreover, we use Adam optimizer (Kingma & Ba, 2015) with the default learning rate 0.001. On MNIST and F-MNIST we restrict the input to be in the range $[0, 1]$. On the CIFAR-10 dataset, following Croce et al., we apply random crops and random mirroring of the images as data augmentation. We also apply a warm-up training schedule for both regularization weights where we linearly increase them from $\lambda/10$ to $\lambda$ during the first 10 epochs. For each dataset and architecture, we perform a grid search over the regularization weights. All the reported models and

the final set of hyperparameters will be made available on acceptance. In order to make a comparison to the robust training of Croce et al.; Wong & Kolter (2018) we take their publicly available models.

We perform adversarial training using the PGD attack of Madry et al. (2018) with a random selection of 50% clean and 50% adversarial examples in every batch. For the $\ell_2$-norm, we used the implementation from Rauber et al. (2017) to perform PGD-based $\ell_2$ attacks. We use the same $\ell_2$-bound on the perturbation as the $\epsilon$ used in robust error. During training, we perform 40 iterations of the PGD attack for MNIST and F-MNIST, and 7 for CIFAR-10. During evaluation, we use 40 iterations for all datasets. Following Croce et al., the step size is selected as $\epsilon$ divided by the number of iterations and multiplied by 2.

### C.1 DETAILS OF STATE-OF-THE-ART METHODS

We use publicly available implementation for all methods. We denote $\mathcal{L}(\mathcal{D}) = \mathbb{E}_{\mathcal{D}}[\ell(x) + \mathcal{R}(x)]$ to be the objective (e.g. as in Eq. 7.) We briefly describe how each technique we compare to in Table 2 formulates the $\mathcal{R}(x)$ term below:

**Gradient Regularization (TULIP) (Finlay & Oberman, 2019).** The Gradient Regularization term of Finlay & Oberman (2019) is formulated as a norm on the gradient of the loss function with respect to the input:

$$\mathcal{R}(x) = \beta ||\nabla_x \ell(x)||$$

**Locally-Linear Regularization (LLR) (Qin et al., 2019).** Qin et al. (2019) propose to regularize the local linearity of the classifier:

$$\mathcal{R}(x) = \lambda \gamma(\epsilon, x) + \mu ||\delta_{\text{LLR}}^T \nabla_x \ell(x)||$$

where $\gamma(\epsilon, x) = \max_{\delta \in B(x,\epsilon)} g(f, \delta, x)$ and $g(f, \delta, x) = |\ell(x + \delta) - \ell(x) - \delta \nabla_x \ell(x)|$

**Locally-Lipschitz Regularization (TRADES) (Zhang et al., 2019).** Trades is the current empirical state of the art and is motivated by the relationship between robustness and local Lipschitzness.

$$\mathcal{R}(x) = \beta \max_{x' \in B(x,\epsilon)} \ell(x, x')$$

**Curvature Regularization (CURE) (Moosavi-Dezfooli et al., 2019)** Propose to regularize the curvature of the loss with respect to the input:

$$\mathcal{R}(x) = \lambda ||H(x)||_F^2$$

Note that $H$ is estimated through iterative approximation by computing finite differences.

## D ADDITIONAL EXPERIMENTS

### D.1 VISUALIZING THE EFFECTS OF REGULARIZATION

We explore the geometry of polytopes. In Fig. 5, we overlay histograms corresponding to the radii of Dikin ellipsoids of samples pre- and post-robust training. It is clear that robust training increases the size of polytopes, agreeing with the intuition of the GR certificate.

In Fig. 6 we visualize the decision boundary for different values of $\lambda_{\text{L}}$ via Eq. 11. As expected, we see that as this weight is increased, the network imposes higher cost on the local loss curvature resulting in a linearization of the decision boundary and lower overall confidence in predictions over the domain. However, something unusual happens when $\lambda_{\text{L}}$ exceeds a threshold: the network exhibits uncertainty uniformly over the domain, and both clean and adversarial test accuracy drop significantly.

In Table 4, we highlight the importance of both parts of the regularization, i) penalization of the distance to boundary of the polytope ii) penalizing the curvature of decision boundaries across polytopes via ablative experiments. We train FC1 on MNIST using the each component of the regularization scheme. We clearly see that lower bounds on robustness computed using PGD is always significantly better when both terms are used. In order to achieve the best empirical performance, it is necessary to both increase the distance of the points to the boundaries of the polytope *and* regularize the curvature of the decision boundary between polytopes.

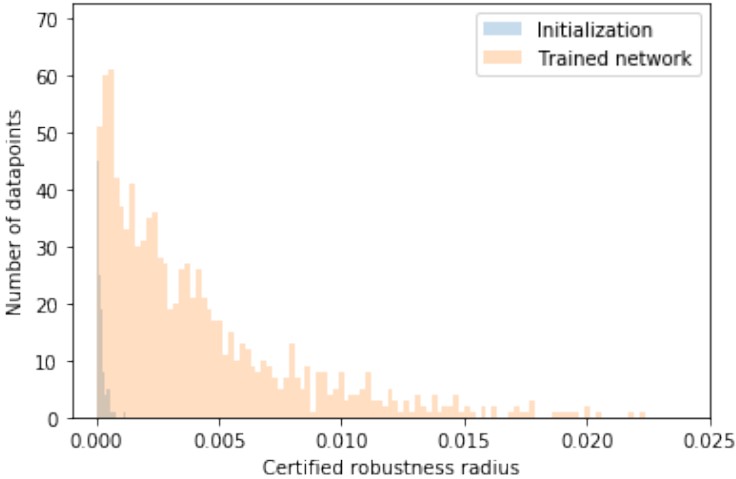

Figure 5: GR over robust training. As expected, the volumes of polytopes (certification of robustness) increases over training.

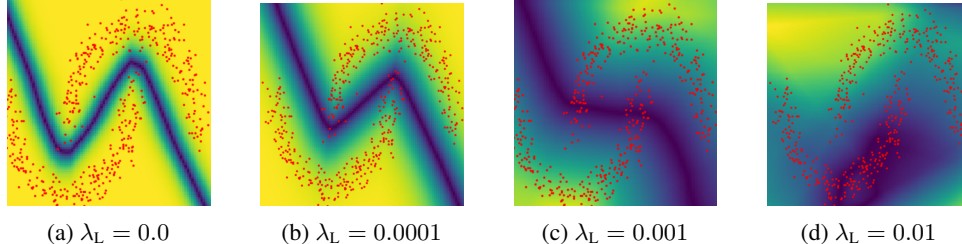

(a) $\lambda_L = 0.0$      (b) $\lambda_L = 0.0001$      (c) $\lambda_L = 0.001$      (d) $\lambda_L = 0.01$

Figure 6: We plot decision boundaries via Eq. 11 for different values of $\lambda_L$. As expected, as $\lambda_L$ increases, the decision boundary exhibits less curvature.

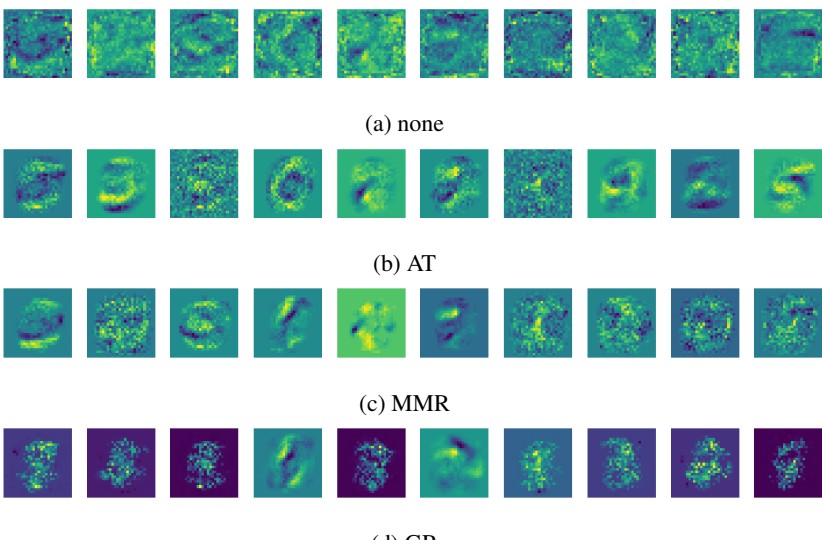

(a) none

(b) AT

(c) MMR

(d) GR

Figure 7: We plot visualization of neuron weights for networks trained with different regularizers on MNIST.

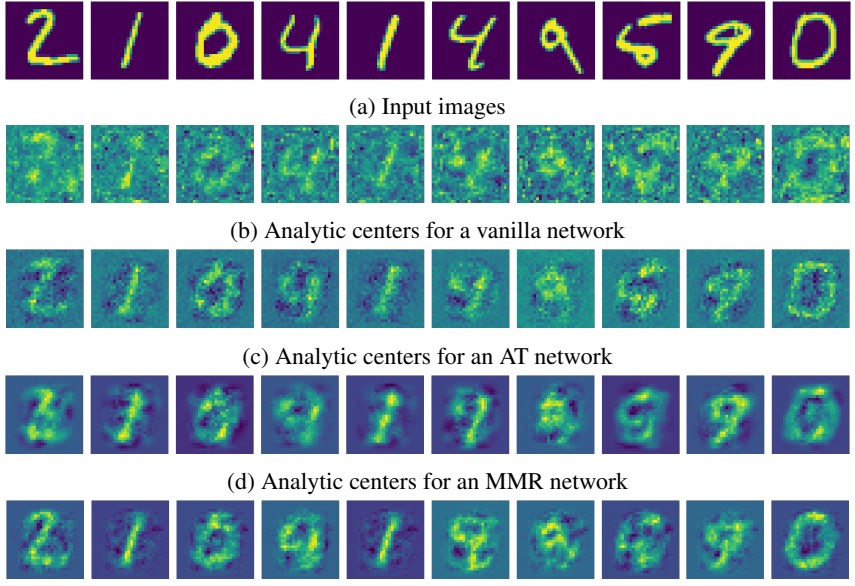

(a) Input images

(b) Analytic centers for a vanilla network

(c) Analytic centers for an AT network

(d) Analytic centers for an MMR network

(e) Analytic centers for a GR network

Figure 8: We plot the analytic centers for trained networks. We see that the analytic centers for vanilla networks roughly correspond to noise. However, the analytic centers of polytopes induced by robust networks are clearly interpretable and similar to the inputs they are conditioned on.

## D.2 ANALYSIS OF ELLIPSOIDS AND ANALYTIC CENTERS

The geometry of the partitioning and the associated polytopes induced by an ReLU network is not obvious. Intuitively, we would like the partitions to be semantically reasonable, i.e. cleanly partition the domain into class-regions. However, this is not the case as implied by the existence of adversarial examples. An immediate consequence of the GR certificate is that we expect "realistic" examples to lie far from the boundary of their respective polytopes. We encourage this via the $J_{\text{ACR}}$ regularizer.

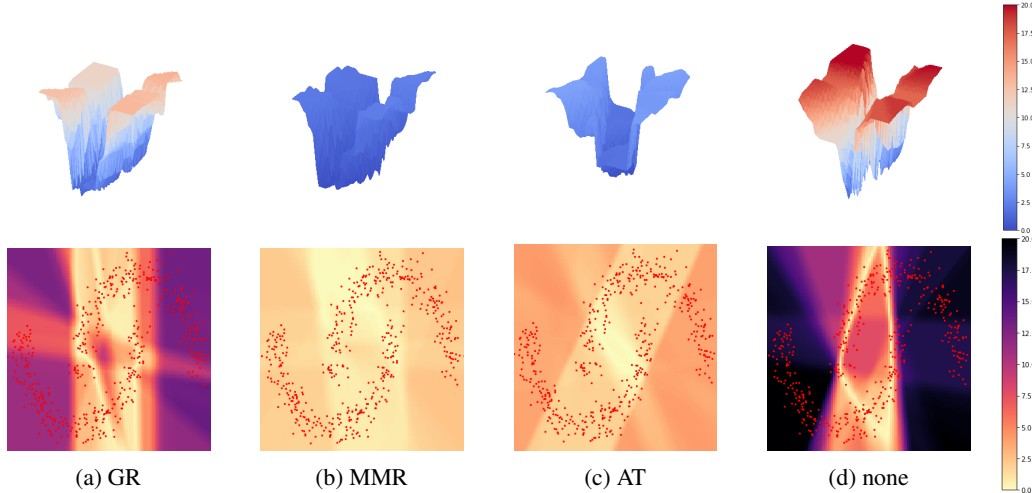

Figure 9: We evaluate the stochastic local Lipschitz constant (Eq. 12) networks trained in different ways on the Two Moons dataset uniformly over the grid. We see that GR exhibits local smoothness in regions around potentially adversarial examples, while preserving nonsmoothness elsewhere. In contrast, the networks trained with MMR and adversarial training exhibit smoothness globally.

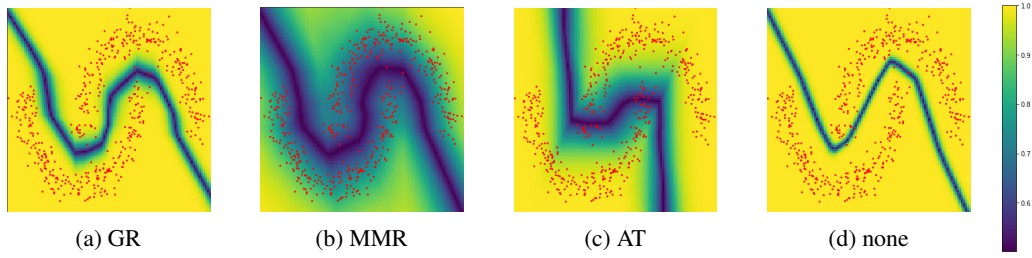

Figure 10: We plot the decision boundaries for robust and non-robust networks on the Two Moons dataset corresponding to Eq. 11 —the difference in likelihoods between the most likely class and the second most likely class. We see that relative to networks trained with other methods, networks trained with GR exhibits a larger gap - i.e. more confidence for regions around the decision boundary ($P_a - P_b$ is large).

As a consequence, after training a robust network, we expect the true analytic centers of a given polytope to be semantically interpretable and close to "real" samples. We demonstrate that this is the case in Figure 8. First, note that the ACs of polytopes produced by a vanilla network are entirely noisy. We hypothesize (Fig. 5) that partitions converge early in training, and that the linear functions on each partition are the focus of the majority of training. As a consequence, the content of polytopes for non-robust networks are near-random. It is clear that the analytic centers of robust network polytopes are more interpretable (Fig. 5).

## D.3 VISUALIZING THE STRUCTURE OF ROBUST MODELS

We demonstrate several properties of networks trained with robust regularization terms. We train a single hidden layer feed forward ReLU network on the Two Moons dataset. The decision boundaries are presented in Fig. 10. We estimate decision boundaries by plotting

$$r(x) = \max_{i \in k} L(y) - \max_{\substack{j \in k \\ j \neq i}} L(y) \tag{11}$$

where $L(x) = \frac{e^x}{\sum_{k \in [K]} e^x}$ and $y$ are the logits of $f$ evaluated at $x$. In other words, $r(x)$ directly corresponds to the second term of the Lipschitz robustness bound of $f$ at $x$. We first note in Fig. 10 that MMR and GR preserve the decision boundary learned by the vanilla network for the most

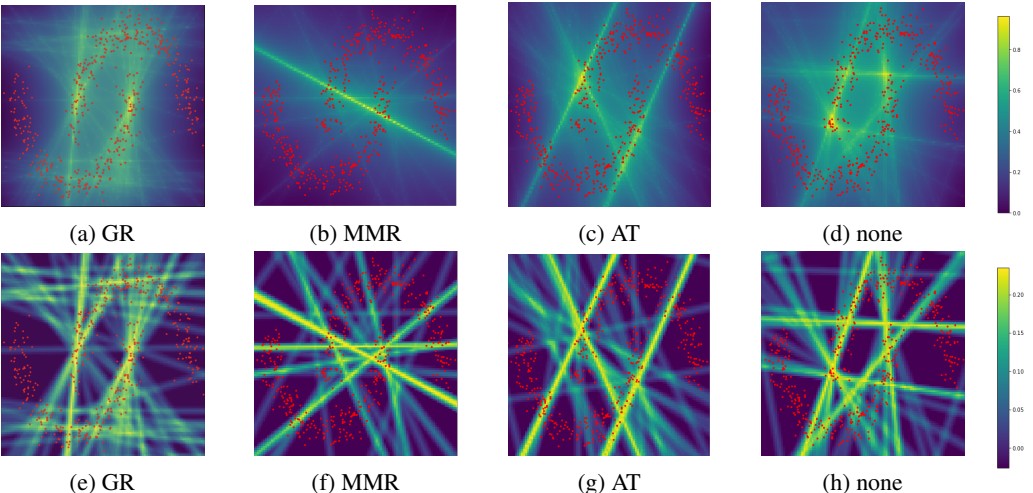

Figure 11: We evaluate the GR (top) and MMR (bottom) regularization terms uniformly over a grid for networks trained in different ways on the Two Moons dataset.

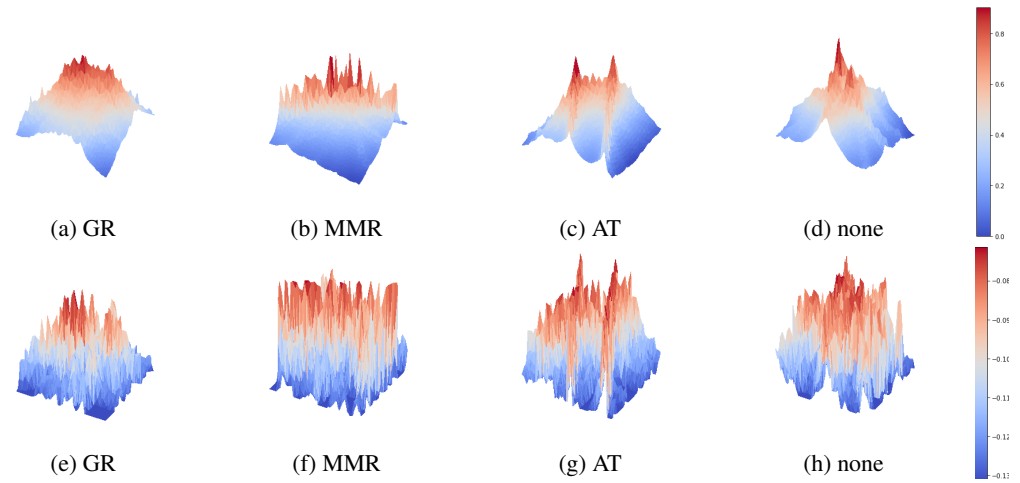

Figure 12: We evaluate the GR (top) and MMR (bottom) regularization terms uniformly over a grid for networks trained in different ways on the Two Moons dataset. Although the numerical scale of the costs differ, it seems clear that GR exhibits the desirable property smoothness over the inputs.

part, however, the network trained via adversarial training has over-regularized, resulting in a sub-optimal classifier. Interestingly, the regularizer we propose does exhibit a significant gap between the most likely and second most likely class likelihoods this is a desirable property with respect to the GR robustness guarantee. We also notice that the network trained with GR exhibits less uniform smoothness compared to the other regularizers as seen in Fig. 9. MMR and adversarial training appear to over-regularize the network by imposing smoothness uniformly over the domain.

In Figures 11 and 12 we demonstrate that the network trained with GR exhibits spatial smoothness in contrast to the MMR term, i.e. neighboring points exhibit similar regularization loss. We hypothesize that this may partially provide evidence for (1) the empirical reduction in the number of linear partitions and increases their size (Fig. 5) such that more points are co-located within the interior of the same polytope and (2) smoother optimization.

In Fig. 9 we visualize the local smoothness of robust and non-robust networks. We uniformly sample points from the 2-$d$ cube and approximate the *stochastic local Lipschitz constant* of $f$ by computing the following term:

$$\hat{L}(x) = \frac{1}{N} \sum_{i=1}^{N} \frac{f(x) - f(x - \delta_i)}{\delta_i^2} \tag{12}$$

where $\delta_i \sim \mathcal{U}_{(0,\epsilon]}$. Note that for large values of $L$ around a point $x$, $f$ exhibits local instability, and potentially brittleness with respect to adversarial perturbations of $x$. In conjunction with Fig 10, we see that GR along with AT and MMR exhibits behavioral properties that are typical of robust networks (e.g. local smoothness and confidence).

We also plot visualizations of layer-weights for a fully connected network It is well-known that structured neuron weights lead to more robust networks (Allen-Zhu & Li, 2020). However it has only been recently deeply explored in the context of adversarial machine learning (Allen-Zhu & Li, 2020). We demonstrate that networks trained with GR exhibit this property of encouraging neurons to learn structured features.

## E EFFICIENT PRUNING OF REDUNDANT POLYTOPE REPRESENTATIONS

To remedy the issue of redundant representations of polytopes, we propose the following masked-version of ACR:

$$ACR_{\text{masked}}(x) = \sum_{i \in \Omega} \log(a_i - \sum_{j=1}^{d} v_{ij} x_j), \tag{13}$$

where $\Omega$ is the index-set of irredundant hyperplanes. To compute $\Omega$, we leverage the primal-dual equivalence between half-plane intersections and convex hulls and adopt the approximate convex-hull algorithm of Sartipizadeh & Vincent (2016) to prune redundant hyperplanes.

Given a finite set of points $S = \{x_1, \ldots, x_N\}$ where each $x_i \in \mathbb{R}^d$, Sartipizadeh & Vincent (2016) propose to iteratively build an estimate of an $\epsilon$-approximate convex hull (denoted by $\mathcal{E}$) by selecting a point to add to $\mathcal{E}$ such that the maximum over all projections from $S \setminus \mathcal{E}$ to $\mathcal{E}$ is minimized (i.e. is an *extreme point* of $S$), where the projection length from a point $x$ to $\mathcal{E}$ is computed by solving the following quadratic program:

$$d(v, \mathcal{E}) = \min_{\alpha_i} ||v - \sum_{i=1}^{|\mathcal{E}|} \alpha_i x_i||_2$$

$$\text{s.t. } \alpha \geq 0, \ \sum_{i=1}^{|\mathcal{E}|} \alpha_i = 1$$

instead of exactly solving this problem, Sartipizadeh & Vincent (2016) propose to select points that are $\epsilon$-close to the convex hull - i.e. by requiring that $\max_{v \in S} d(v, \mathcal{E}) \leq \epsilon$. In summary, at iteration $t$ the algorithm selects $x_t$ to add to $\mathcal{E}$ according to the following decision rule:

$$x_t = \arg\min_{x \in S \setminus \mathcal{E}} \max_{v \in S \setminus \mathcal{E}} d(v, \mathcal{E} \cup x)$$

Note that we are not guaranteed uniqueness, in that there may be multiple $\mathcal{E}$ with the same number of elements, satisfying the distance requirements. However, as $\epsilon$ increases, the size of a minimal representation, $\mathcal{E}$, may decrease.

Notably, the above algorithm runs in time independent-of the dimension: $O(K^{3/2}N^2 \log \frac{K}{\epsilon})$ for $N$ points in dimension $d$, and where $K$ is the number of iterations. Additionally, it is provably correct, i.e. after the algorithm terminates the set $\mathcal{E}$ is guaranteed to be an $\epsilon$-approximate convex hull of $S$ where an $\epsilon$-approximate convex hull is defined to be:

**Definition E.1** ($\epsilon$-approximate convex hull). An $\epsilon$-approximate convex hull of $S$ is the convex hull of a minimal subset $\mathcal{E} \subseteq S$ such that $\forall z \in S, d(z, \mathcal{S}) \leq \epsilon$.

