# OpenReview forum: "Provable Robustness by Geometric Regularization of ReLU Networks"
_ICLR.cc/2021/Conference — Reject_

### Official Review · AnonReviewer4 · 2020-10-27
**detailed theoretical analysis missing**

**Rating:** 4
**Confidence:** 3

**Review:**

As a reader and a reviewer, I am not unable to follow the main theoretical point in the paper. The proof of Theorem 1 is not properly explained with ambiguous reference to other articles.

More specifically, regarding the proof of Theorem 1 in Appendix A.2, the authors do not provide enough details nor explanation for the second part of the statement. Merely writing something like 'follow from papers xxx' without any details makes the proof unreadable.

I think as a basic principle, the authors should consider readers' convenience and provide a self-contained proof and intuitive explanation of the claims. At least they should specify what results from outside are used in their theoretical analysis.

The careless treatment of theory makes the manuscript unreadable, and thus the present form is not qualify for an ICLR submission.

---

> ### Author Response · Authors · 2020-11-23
> **Response to R4**
>
> We thank the reviewer for their constructive comments. To address the issue brought up, we have rewritten the derivations of Theorem 1 and its associated Lemma in the appendix. We have also clarified and rewritten several elements of the theorem in the paper and its discussion. We have composed a preliminaries section in the appendix and have provided the curvature bounds presented in Moosavi-Dezfooli et al. For completeness, we also have included a proof of a fact stated in the paper - that the Dikin Ellipse is contained within its associated polytope. We hope that these additional clarifications and details help clear up any ambiguities.
>
> To add additional context, the existing work on defences to adversarial attacks fall into two categories: (1) heuristic methods (e.g. adversarial training) (2) techniques that optimize a provable bound on the “robust radius” given an input (e.g. the method proposed in our work). We introduce such a (novel) bound on the robust radius given an input in section 3.4 (the Dikin Certificate) and propose a simple, geometrically motivated regularization term (ACR), which when optimized, improves the bound. Theorem 1 and its proof serve as formal statements of (1) the bound and (2) the fact that our regularizer maximizes this bound under weak assumptions on the polytope geometry. We also provide experiments to show the regularization works in practice - and works generally to induce models that are simultaneously robust to multiple perturbations (namely, l-2 and l-inf perturbations). We have added additional clarification in the new draft.

---

### Official Review · AnonReviewer2 · 2020-10-27
**some questions on the experimental results**

**Rating:** 6
**Confidence:** 3

**Review:**

The paper present some regularization schemes for ReLU networks, based in geometrical properties (polytopes and analytical centers), aiming at robustifying networks against adversarial examples. The idea is to train the network such that the partition of the input space contains many less linear regions, and such that in each region, training points are gathered around the analytical center (and thus far from boundaries), making more difficult the attack task.
Doing so the authors can  give certificates for robustness.  The proposed approach appears to be competitive in terms of bounds and do so with less hyper-parameters than other methods.

The idea of using regularization terms to make networks more robust makes sense, the particular proposed regularizers have some quite easy to catch motivation.
The presented results match state-of-the-art, however I have some (maybe naive) questions :
-> why TE are missing sometimes?
-> how can LB>UB (CNN l_inf) ? I would have say that max(LB)=min(UB)=true minimal distortion ?
Fig 3 is unreadable which is a pity since I think it could be some relevant information, its interpretation is quite brief too.

Overall I think the presented work is interesting and quite well executed. Theoretical part seems ok, but I'm not familiar with some geometric notions so I might have missed things.
I tend to accept the paper but I'm waiting for more explanations on the experimental results.

---

> ### Author Response · Authors · 2020-11-23
> **Response to R2**
>
> We thank the reviewer very much for their positive comments. Following the other reviewers’ suggestions, we have added more experiments as stated in the revision summary and clarified several points regarding the bound and regularization term. To respond to your questions:
>
> 1. Missing test errors: Note that unlike other reported results, we demonstrate empirical robustness against multiple perturbation types by training one model per architecture and dataset. The other models compared to are tuned per-perturbation. Thus, we forgo duplicating the numbers for clean test error for variations in perturbation for the same dataset since the clean TE only needs to be reported once (i.e. the clean test-error is the same regardless of perturbation type). Regarding blocked out grey results for CIFAR 10 under the FC1 heading, we do not report these results as fully connected networks perform very poorly in general on CIFAR 10. We have reformatted the table (and switched the order of the CNN/FCNN columns) to make this more clear.
>
> 2. LB>UB (CNN l_inf): We thank the reviewer for this comment. We note that the adversarial lower-bound computed in the paper is computed over the entire test-set, while the adversarial upper bound (and the average robust radius) is computed on the first 1000 points. We have clarified this in the paper and intend on normalizing the results in the final draft.
>
> 3. Fig 3: We highlight that despite our provable bound being a lower-bound on the MMR bound, that our empirical bounds match MMR quite closely. We have added additional clarification in the new version and increased the size of the plots.

---

### Official Review · AnonReviewer3 · 2020-10-29
**An slight improvement over an existing regularization method for neural network robustness, but more convincing experiments needed.**

**Rating:** 5
**Confidence:** 4

**Review:**

The authors exploit the piecewise linear nature of ReLU neural networks to design a new regularizer that improves the robustness of the neural network. It can be viewed as a alternative to the regularizer proposed in Croce et al. (2018) -- the current regularizer uses the analytic center, whereas the previous work (named MMR) guarantees that are straightforward consequences of the geometric interpretation. The experimental setup is similar to the one in Croce et al. (2018), which compares the result of different adversarial defense methods on MNIST, F-MNIST, and CIFAR10 on small shallow networks. These computational results generally slightly outperform MMR on MNIST.

The paper is polished and well-written, and the use of the analytic center is reasonably motivated via the geometry of the neural network.

However, the paper has a few major drawbacks that need to be addressed before acceptance.

(1) Novelty: The paper can be seen as a twist on Croce et al., working off some of the same intuition and using very similar experiments. The theoretical results are a simple consequence of using the analytic center. Hence, this work can be viewed as incremental.

(2) Experimental Results: Given that the set of regularization approaches has significantly grown since the prior paper was published, the authors should at least compare their methods with some of these. This is especially important given that the results are not much better than MMR (and worse in some cases).

(3) Computational Considerations: The computation effort needed to incorporate this regularizer should be discussed explicitly so the reader can understand the trade-offs between incorporating this regularizer vs. other defense methods.

Side note:
The authors should update the citation for Croce et al. to refer to the conference version of the paper.

---

> ### Author Response · Authors · 2020-11-23
> **Response to R3**
>
> We thank the reviewer for their constructive comments. To address the issues mentioned:
>
> 1. Novelty: We agree that this work is a follow-up to Croce et al., however, we argue that our empirical results demonstrate significant advantages of our algorithm - namely fewer hyperparameters (i.e. by taking into account the geometry of the polytope, we avoid having to tune heuristic parameters, such as K in the MMR formulation), robustness to gradient obfuscation, adaptation to the local polytope geometry of samples, and efficiency of optimization. We have included a section (4.2) highlighting an additional advantage of using the LDR component of our regularizer - namely, robustness to gradient obfuscation. We show that MMR-based regularizers appear to suffer from this issue while GR does not.
>
> 2. Experimental results: We agree with the reviewer. We have added additional experiments to the main text (Table 2) comparing our work to recent, state-of-the art first and second-order regularization-based algorithms (TRADES, LLR, TULIP, CURE). In summary, our method demonstrates competitive performance with the provable approaches (matching or exceeding performance with respect to robust accuracy), although the heuristic methods remain state-of-the-art.
>
> 3. Computational considerations: Computing both the regularizer and certificate requires negligible overhead compared to related methods. Note that given $x$, a forward pass through $f$ is sufficient to compute both the linearization of $f$ around $x$ and the associated region (the polytope). Given these entries, optimizing the regularization term corresponds to solving a smooth, unconstrained convex optimization problem - which can be done very efficiently using gradient descent. Computing the Dikin certificate involves computing the polytope and performing an eigenvalue decomposition on the Hessian (a symmetric, PSD matrix) of the Log-barrier function (note, this is just to compute the certificate - we don’t do this during training). Alternatively, recent state-of-the-art regularization methods (CURE, LLR, TULIP, TRADES) typically necessitate sampling, and/or computing gradients (to recover adversarial perturbations) on individual samples/batches - this incurs a significant overhead during training.
>
> We also thank the reviewer for catching issues with the citation of Croce et. al.. We have fixed this in the most recent version.

---

### Public Comment · ~Thiago_Serra1 · 2020-11-12
**Connection with linear regions**

As someone who studies the linear regions of neural networks, I appreciate whenever I see a verification paper discussing that.

You mention a few papers on that topic, which are definitely important but do not cover everything that has been done. Hence, in case the authors are not aware, I would like to bring to your attention a work that some colleagues and I published at ICML 2018, which sheds a light on the depth vs. width trade-off on the number of linear regions defined by rectifier networks:

https://arxiv.org/abs/1711.02114

---

> ### Author Response · Authors · 2020-11-23
> **Thank you for the comment!**
>
> We appreciate the interest. Actually, we are familiar with the paper you link to, and are happy to include a reference to it as suggested. It would be interesting to explore the relationship between the exact/empirical number of linear regions and robust regularizers.

---

### Author Response · Authors · 2020-11-23
**To all reviewers**

We thank the reviewers for their constructive comments. We have uploaded a new draft which clarifies questions and addresses issues brought up by all reviewers.

- We have re-written all derivations in the appendix according to R4’s comments.
- We have included a new section (4.2) clarifying an advantage of using the LDR component of our regularizer - namely, robustness to gradient obfuscation. We show that MMR-based regularizers appear to suffer from this issue.
- We have also added additional experiments (Table 2) comparing our work to recent, state-of-the art first and second-order provable and heuristic regularization-based algorithms (TRADES, LLR, TULIP, CURE). We demonstrate that our method recovers comparative performance to other provable methods with less overhead and fewer hyperparameters.

We would also like to clarify for R2 that unlike MMR, which performs a grid-search for each individual attack-type, we apply the same model when evaluating performance on l-2 and l-inf attacks. Also, we have made a few changes for additional polish: In addition to revising several unclear sections (e.g. adding a sentence discussing computational overhead in Section 3.3), we have switched the order of the columns in Table 1 for clarity, reproduced Figure 11, and increased the size of Figure 3. We have also made some very minor edits to the code submission.

---

### Decision · Program_Chairs · 2021-01-07
**Final Decision**

**Decision:**

Reject

**Comment:**

The paper presents a novel regularization scheme for showing provable robustness guarantees for the class of ReLU networks.
The reviews of this paper were mixed but leaning more to reject. Even though geometric approaches to adversarial robustness are appealing there are too many issues left open (see below for detailed comments) so that the paper is not above the bar for ICLR.

Detailed comments:
 - incremental resp. small theoretical contribution: the bounds in Theorem 1 are worse than what has been derived in Croce et al (2019), which is admitted only in the Experiments section but this should directly be discussed after Theorem 1, or are based on other's work Moosavi-Dezfooli et al(2019)
 - the proofs are still not very clear even after the update e.g. it is not clear to me what is meant with the distance to the decision boundary d_D(x) as this quantity cannot be computed (this is the whole point here)
 - the experimental results are in most of the cases worse than prior work (KW and MMR). The authors argue that their results hold for both threat models but then the l_infty/l_2 robustness of the other models should have been provided to make the point.
Moreover, if multiple norm ball threat models is the point, then the authors should compare to the follow up work of Croce et al at ICLR 2020 which explicitly optimizes all l_p-balls.
- the point with gradient obfuscation of MMR does not make sense to me as the gap between upper and lower bounds on the robust error is quite small for MMR in most cases. Gradient obfuscation would mean that the PGD attack fails and thus the lower bound would have to be close to zero.
- Figure 1c) looks like the Figure from the book of Boyd and Vandenberghe - if this is the case then it has to be correctly referenced in the caption